# Upcycling of groundwater treatment sludge to magnetic Fe/Mn-bearing nanorod for chromate adsorption from wastewater treatment

Zhan Qu[1], Wenqing Dong[1], Yu Chen[2], Ge Dong[1], Suiyi Zhu[1]*, Yang Yu[3], Dejun Bian[1]

1 School of Environment, Northeast Normal University, Changchun, China, 2 Jilin Institute of Forestry Survey and Design, Changchun, China, 3 School of Chemical Science and Engineering, Longdong University, Qingyang, China

* papermanuscript@126.com

**Data Availability Statement:** All relevant data are within the manuscript and its Supporting Information files.

## Abstract

Groundwater treatment sludge is a Fe/Mn-bearing waste that is mass produced in groundwater treatment plant. In this study, sludge was converted to a magnetic adsorbent (MA) by adding ascorbate. The sludge was weakly magnetised in the amorphous form with Fe and Mn contents of 28.8% and 8.1%, respectively. After hydrothermal treatment, Fe/Mn oxides in the sludge was recrystallised to siderite and rhodochrosite, with jacobsite as the intermediate in the presence of ascorbate. With an increment in ascorbate dosage, the obtained magnetic adsorbent had a significant increase in chromate adsorption but a decrease in magnetisation. When the $M_{ascorbate}/M_{Fe}$ molar ratio was 10, the produced MA-10 was a dumbbell-shaped nanorod with a length of 2–5 μm and a diameter of 0.5–1 μm. This MA-10 showed 183.2 mg/g of chromate adsorption capacity and 2.81 emu/g of magnetisation. The mechanism of chromate adsorption was surface coprecipitation of the generated $Cr^{3+}$ and $Fe^{3+}/Mn^{4+}$ from redox reaction between chromate and siderite/rhodochrosite on MA-10, separately. This study demonstrated an efficient recycling route of waste sludge from groundwater treatment to produce MA for treating chromate-bearing wastewater.

## 1. Introduction

Chromate-containing wastewater, which needs to be effectively treated before discharging due to the high physiological toxicity of chromate to plants and animals, is widely produced in smelting and tannery factories [1, 2]. To prevent pollution, the Chinese government has reduced the maximum discharging concentration of chromate to 0.1 mg/L [2]. Many strategies, such as chemical precipitation [3], ultrafiltration [4] and ion exchange and adsorption [2], have also been applied to remove chromate from wastewater. Among these strategies, adsorption is considered as an economic and feasible method in treating chromate-containing wastewater. Industrial wastes, such as iron sludge from groundwater treatment [5], fly ash from coal combustion [6] and red mud from alumina refining [7], have been used as low-cost

**Funding:** This work was partially funded by the National Natural Science Foundation of China (Grant Nos. 51578118, 51678273, 51878134, and 51878133), the Fundamental Research Funds for the Central Universities (Grant No. 2412017QD021).

**Competing interests:** The authors have declared that no competing interests exist.

adsorbents for direct adsorption of chromate. However, after adsorption, separation of industrial wastes commonly consists of complicated centrifugation and tedious coagulation, which become problematic in wastewater treatment. When the magnetic species was incorporated into the industrial wastes, it conferred magnetic response on the wastes, so that the wastes can be easily separately from water in a magnetic field [8, 9]. Thus, these wastes could be converted to magnetic adsorbent, which favors the wastes' separation and reduces the size of clarifier accordingly [10].

Groundwater treatment sludge is the precipitate of backwash wastewater in groundwater treatment plant. In a previous research, approximately 1 t of sludge was produced when treating 5000 t of groundwater [11]. The produced sludge comprised ferrihydrite, hematite and impurities such as Si/Al oxides [9, 12]. The ferrihydrite in the sludge was 16.6–33.7 wt.% [13, 14], and it had a special structure wherein each iron atom was covalent with six oxygen/hydroxyl clusters [15]. Thus, hydrogen groups were abundantly available on ferrihydrite surface for chromate coordination [11, 16]. In addition, the ferrihydrite in the sludge could be hydrothermally transformed to maghemite and magnetite [12, 13] for the converted sludge to have good magnetic response and to be easily collected by a magnet after use. During ferrihydrite conversion, the hydroxyl groups on ferrihydrite surface exhibited coordinated unsaturation via dihydroxylation [17]; thus, small ferrihydrites aggregated to generate aggregated maghemite and/or hematite [18]. The covalent hydrogen groups per iron atom decreased after hydrothermal treatment [16]. The adsorption capacity of chromate on the converted sludge lowered in comparison with that of the raw sludge. On this basis, adsorption capacity needs to be improved with a feasible approach.

In this study, sludge was in situ conversed to magnetic adsorbents (MAs). Unlike the conventional adsorbent with abundant surface hydroxyl groups for chromate coordination [3], the produced MAs were rich in siderite and rhodochrosite. The produced MAs exhibited high chromate adsorption via a combined effect of a redox reaction between chromate and the two carbonate minerals and a surface precipitation reaction of the generated $Cr^{3+}$ and $Fe^{3+}/Mn^{4+}$ cations.

## 2. Materials and methods

### 2.1 Ethics statement

We got full permission from Northeast Normal University school of environment, conduct research on this topic in 137 laboratory and the geographic coordinates is 125.43° E, 43.83° N.

### 2.2 Groundwater treatment sludge pretreatment

Groundwater treatment sludge was discharged from Kulunyin potable water plant located at Inner Mongolia, China. The sludge was sampled and then vacuum-dried at 55°C for 36 h before characterisation by X-ray fluorescence spectroscopy (S4-Explorer, Bruker, XRF, Germany). The major composition of sludge was Fe (28.8%), Mn (8.1%), Si (8.1%), Al (2.3%), Ca (2.1%) and Mg (0.5%).

### 2.3 Synthesis of magnetic adsorbent

Hydrothermal treatment of the sludge was conducted as follows. Ascorbate at the $M_{ascorbate}/M_{Fe}$ molar ratio (short for molar ratio) of 1 was mixed with 0.7 g sludge in 30 mL 0.35 M NaOH solution. After stirring at 120 rpm for 10 min, the mixture solution was dumped in 50 mL Telfon vessel, heated at 160 °C for 5 h and then water-cooled down to below 25°C. The brownish particles were generated in the vessel, collected and washed three times with

deionised water, followed by vacuum-drying at 55 °C for 36 h. The obtained magnetic product was denoted as MA-1. The reference experiment was also conducted by varying the molar ratio from 1 to 10, and the corresponding product was named as MA-10.

## 2.4 Adsorption experiments

MA-1 and MA-10 were used for chromate adsorption as follows. The stock solution containing 10 mg/L chromate was adjusted to pH 4 with 1.5 M HCl. In the adsorption experiment, MA-1 and 20 mL stock solution was mixed in a series of 50 mL conical flask, sealed and shaken at 120 rpm. At the given time, a flask was sampled and magnetically treated to separate MA-1. The chromate in the residual solution was determined using inductive coupled plasma–optical secretion spectrometry (Avio-200, ICP-OES, USA, PerkinElmer). In parallel, the adsorption kinetics of MA-10 for chromate was also investigated following the adsorption procedures of MA-1. Batch experiments of chromate adsorption on MA-1 and MA-10 were performed at a chromate concentration of 0–1000 mg/L and an equilibration time of 24 h. Each experiment was performed in triple, and average data were reported.

## 2.5 Characterisation of the sludge and adsorbents

The sludge and the two MAs before and after chromate adsorption were characterised by SEM, XRD, XRF, XPS and Mössbauer spectroscopy. The related method was described in the supplementary files.

# 3. Results and discussion

## 3.1 Transformation of ferrihydrite in the sludge

The composition of sludge, MA-1 and MA-10 was determined by X-ray fluorescence spectroscopy (S4-Explorer, Bruker, XRF, Germany). After hydrothermal treatment, the product MA-1, prepared at $M_{ascorbate}/M_{Fe}$ molar ratio (short for molar ratio) of 1, showed a high Fe/Mn content (34.2% and 9.6%, separately) and a low Si/Al content (4.5% and 1.1, separately) (Fig 1), in comparison with the raw sludge, due to the dissolution of Si/Al oxides (e.g. kaolinite) under alkaline condition (Fig 5A) with the release of $Si(OH)_4^-$ (Fig 5B) and $Al(OH)_4$ to the solution [19]. However, when the molar ratio was increased to 10, the Fe and Mn in product MA-10, were 25.4 and 7.1 wt.% (Fig 1), apparently lower than those in the raw sludge and MA-1, which were assigned to the reductive dissolution of Fe/Mn at neutral condition (Fig 5A and 5B). But the Si and Al in MA-10 were 10.8 and 3.9 wt.%, higher than those in the raw sludge and MA-1, demonstrating that the release of Si/Al to solution was retarded with the solution pH decreasing from 12.1 to 7 (Fig 5A).

The crystal phase of ferrihydrite in the sludge was characterised by XRD and Mössbauer (Figs 2 and 3). The sludge exhibited the typical peaks of hematite (JCPDS 33–0664) and Si/Al oxides, e.g. quartz, dmisteinbergite and kaolinite. Ferrihydrite in the sludge was weakly crystallised and recorded in Mössbauer spectra (Fig 3). The relative area of ferrihydrite in the Fe oxides of sludge were 67.9% (Table 1), indicating the abundance of ferrihydrite in the sludge.

At the same time, zeta potential test was conducted on the original iron mud and hydrothermal reaction products. The results showed that the zeta potential of the original iron mud changes from 7.5 mV to −18.5 mV (MA-1) and −39.6 mV (MA-10), thereby proving that the surface of hydrothermal reaction product has negative charge. In an aqueous system, the surface of ferrihydrite is covered with -FeOH groups [20].

For MA-1, the peaks of the hematite at 33.1° and 35.6° intensified, and two new peaks belonging to jacobsite (JCPDS 10–0319) at 2θ = 29.8° and 35.1° appeared (Fig 2). The relative

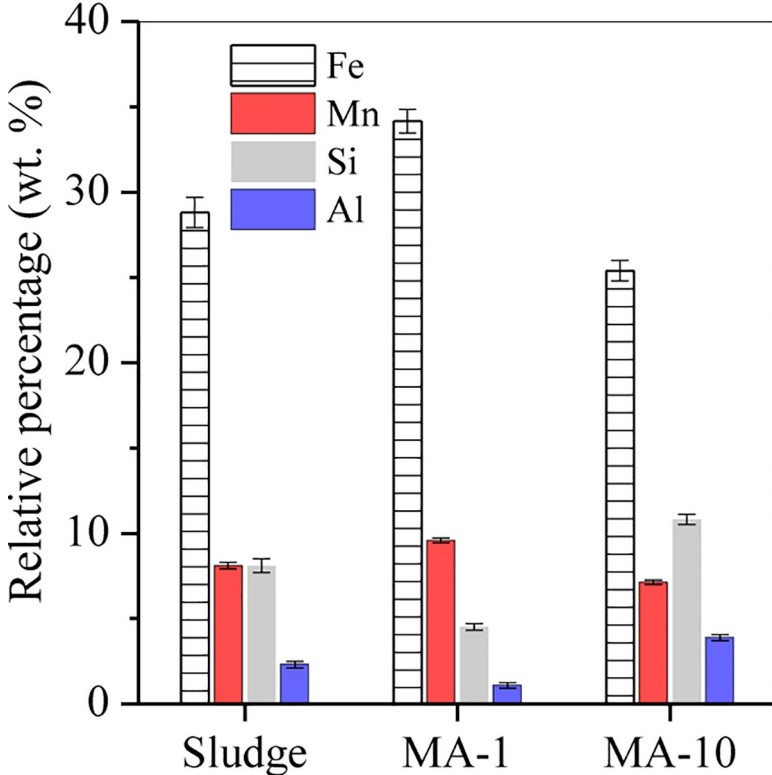

**Fig 1. Relative percentage of Fe, Mn, Si and Al in the sludge, MA-1 and MA-10.**

area of the ferrihydrite decreased by 11.2% (Fig 3 and Table 1), suggesting that ferrihydrite was transformed into hematite and jacobsite. In comparison with MA-1, MA-10 showed that the intensity of the jacobsite peaks decreased. Hence, the jacobsite was reduced with the increase in molar ratio from 1 to 10. However, new peaks were observed in MA-10 curve (Fig 2): two peaks belonged to siderite (JCPDS 29–0696) at $2\theta = 24.8°$ and $32°$, whereas the other two peaks corresponded to rhodochrosite (JCPDS 44–1472) at $2\theta = 31.4°$ and $37.5°$. The relative percentage of the siderite increased by 31.6% after hydrothermal treatment. By contrast, the relative area of the ferrihydrite decreased from 56.7% to 34.6% (Fig 3 and Table 1). The results indicated the conversion of jacobsite and ferrihydrite into siderite by overdosed ascorbate.

To investigate the formation of rhodochrosite, the conversion of Mn oxides was also examined by XPS in the hydrothermal treatment of sludge. As shown in Fig 4, the sludge showed a peak at binding energy of 642 eV, which was related to $Mn^{4+}$ in $MnO_2$ [21, 22]. By adding ascorbate, a new peak at binding energy of 640.5 eV belonged to $Mn^{2+}$ in Mn-O bond [22] was observed in MA-1 and MA-10. On this basis, $MnO_2$ in the sludge was reduced by adding ascorbate to generate $Mn^{2+}$-containing oxides, e.g. jacobsite and rhodochrosite.

Fe/Mn oxides in the sludge included hematite, ferrihydrite and $MnO_2$. Among these oxides, ferrihydrite was weakly crystallised and easily transformed to well-crystallised hematite via dehydration between two adjacent surface Fe-O-H groups of ferrihydrite in the alkali hydrothermal conditions [18, 23]. However, the transformation was impeded by adding ascorbate. The introduced ascorbate spontaneously reacted with Fe/Mn oxides to generate free radicles in the presence of dissolved oxygen [24]. Meanwhile, Fe/Mn oxides on the sludge surface was reduced by adding ascorbate with generation of $Fe^{2+}$ and $Mn^{2+}$ (Fig 5(B)). When the generated $Fe^{2+}$ was coordinated to Mn oxides, it was reoxidised and then involved in the formation of

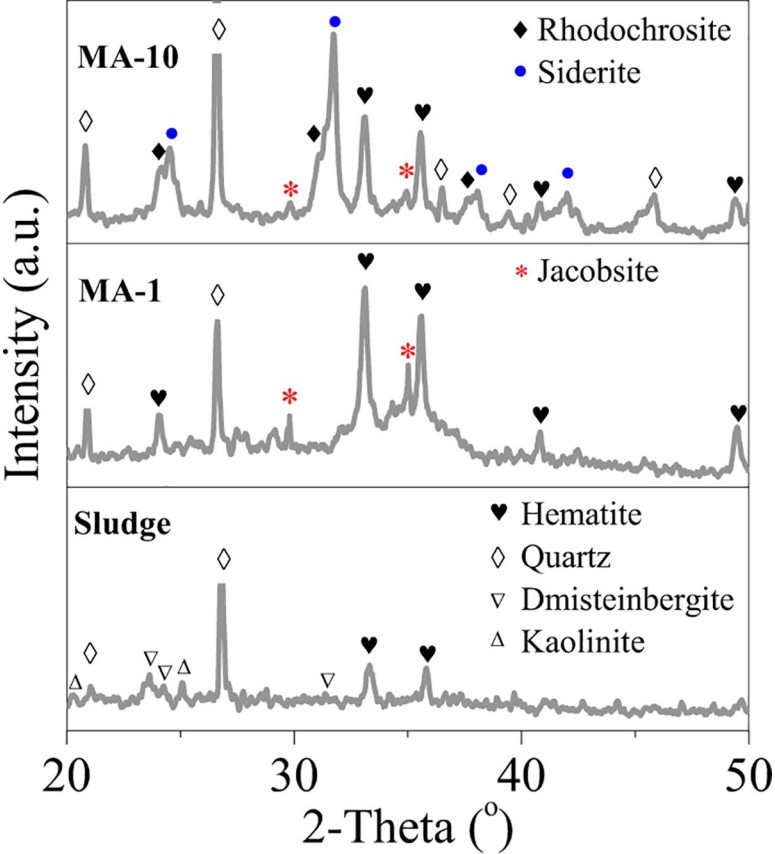

**Fig 2. XRD analysis of the sludge, MA-1 and MA-10.**

$MnFe_2O_4$ [25]. After ascorbate was exhausted, the oxidation of residual $Fe^{2+}$ continued to generate $Fe^{3+}$. In turn, the generated $Fe^{3+}$ was coprecipitated with reduced $Mn^{2+}$ under alkaline condition, resulting in $MnFe_2O_4$ formation [26]. In addition, $Fe^{3+}$ was residual and spontaneously hydrolysed to Fe oxyhydroxide. In turn, the Fe oxyhydroxide covered the formed $MnFe_2O_4$ and blocked the oxidisation of $Mn^{2+}$. In the reaction between ascorbate and Fe/Mn oxides, ascorbate was initially oxidised to L-diketogulonate and further to L-threonate, oxalate. Finally, it decomposed to $CO_2$ and $H_2O$ [27]. As a result, $CO_3^{2-}$ in the solution accumulated with the increase in molar ratio from 1 to 10 (Fig 5(A)).

When the molar ratio was 10, the ascorbate was overdosed to exhaust the dissolved oxygen completely. Then, Fe/Mn oxides were reduced with the generation of $Fe^{2+}/Mn^{2+}$ (Fig 5(B)). These oxides were reacted with carbonate to form siderite and rhodochrosite, separately. In addition, the peaks of dmisteinbergite and kaolinite were not observed after hydrothermal treatment. Meanwhile, the peaks of quartz at $2\theta = 20.8°$ intensified for both MAs (Fig 2). Thus, quartz was recrystallised from Si-containing minerals, such as dmisteinbergite and kaolinite.

## 3.2. Magnetisation

Jacobsite is typically a magnetic species [26]. In this study, the formation of jacobsite in MAs was demonstrated by significant changes in magnetisation. These changes was examined with a magnetometer. As shown in Fig 6, the sludge demonstrated weak magnetism; after hydrothermal treatment, the magnetism significantly increased due to the conversion of Fe/Mn

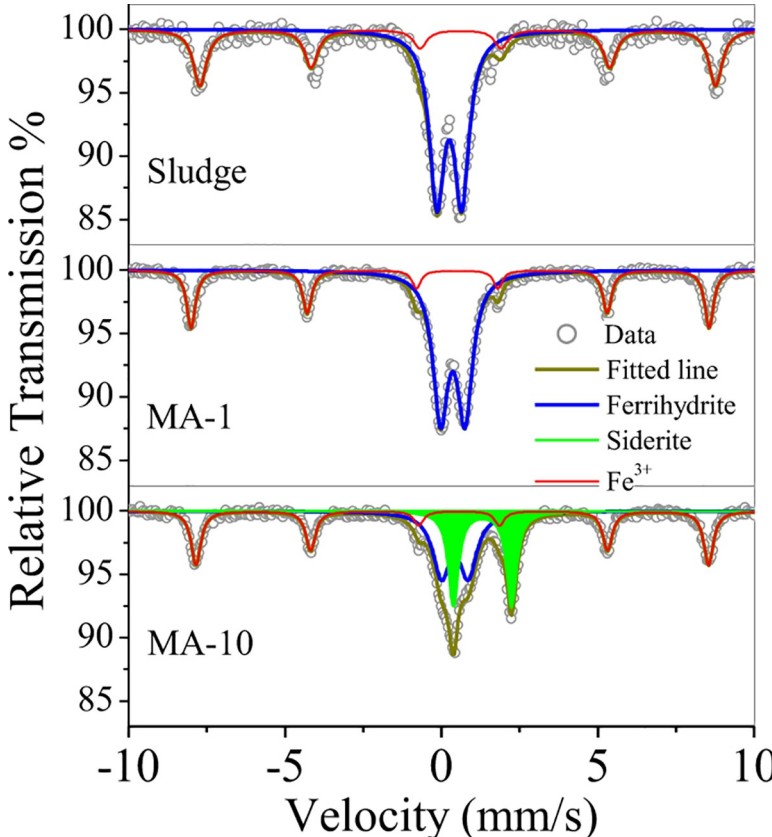

**Fig 3. Mössbauer curves of the sludge, MA-1 and MA-10.**

oxides to jacobsite. However, with the molar ratio increasing from 1 to 10, the saturation magnetisation decreased from 6.7 emu/g of MA-1 to 2.8 emu/g of MA-10. This result was consistent with the abundance of jacobsite in MAs, as shown in Fig 2.

### 3.3. Morphology changes

The sludge demonstrated amorphous aggregates (Fig 7(A)) with uniform distribution of Fe and Mn and dotted distribution of Si. After hydrothermal treatment, the amorphous aggerates of MA-1 grew in size (Fig 7(B)). Si in MA-1 distributed steadily, following theory of dissolution and recrystallisation of Si-containing compounds in the sludge [9]. In comparison with MA-1, MA-10 was a dumbbell-shaped nanorod with a length of 2–5 μm and a diameter of

**Table 1. Mössbauer parameters of the sludge, MA-1 and MA-10.**

| Sample | Component | Isomer shift (mm/s) | Quadruple split (mm/s) | Hyperfine field (KOe) | Relative absorption area (%) |
|---|---|---|---|---|---|
| Sludge | Ferrihydrite | 0.26 | 0.72 | | 67.9 |
| | $Fe^{3+}$ | 0.28 | 0.23 | 509.9 | 32.1 |
| MA-1 | Ferrihydrite | 0.23 | 0.78 | | 56.7 |
| | $Fe^{3+}$ | 0.26 | 0.23 | 513.3 | 43.3 |
| MA-10 | Ferrihydrite | 0.3 | 0.86 | | 34.6 |
| | Siderite | 1.18 | 1.86 | | 31.6 |
| | $Fe^{3+}$ | 0.32 | 0.22 | 507.9 | 33.8 |

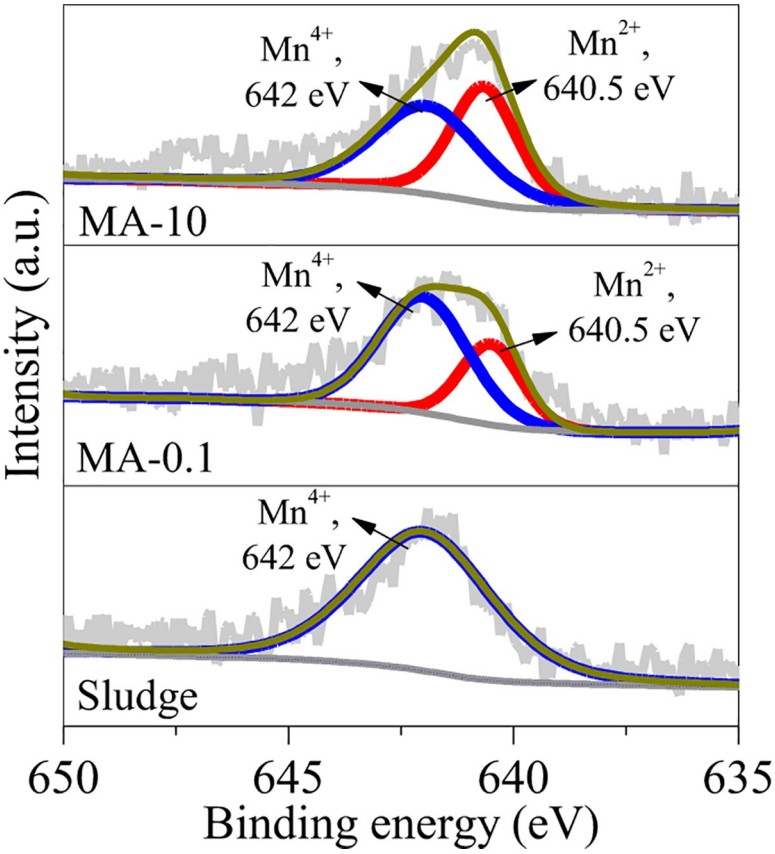

**Fig 4. XPS curves of the sludge, MA-1 and MA-10.**

0.5–1 μm (Fig 7(C) and 7(D)), thereby corresponding to the formation of siderite and rhodo-chrosite. Moreover, element C was not observed in the sludge and MA-1 but observed in MA-10 due to the formation of carbonate minerals, e.g. siderite and rhodochrosite.

## 3.4. Chromate adsorption

As a toxic species in smelting and tannery wastewater, chromate was targeted for adsorption by MA-1 and MA-10 in this study (Fig 8). The adsorption data of chromate on MA-1 and

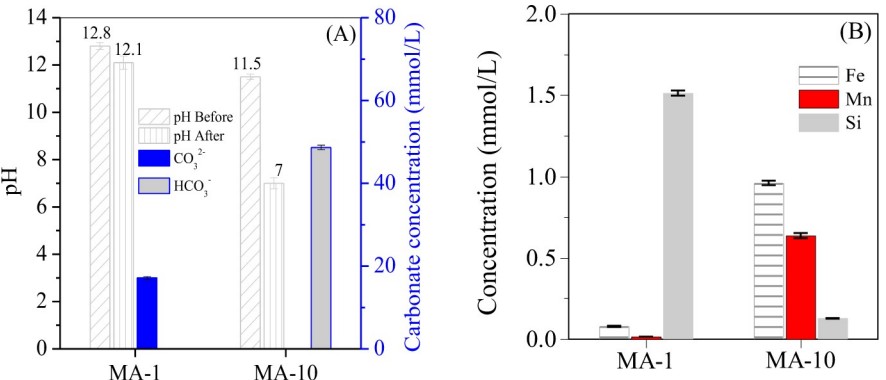

**Fig 5.** (A) pH value of the solution before and after hydrothermal treatment and the carbonate concentration after hydrothermal reaction and (B) Fe, Mn, Al and Si concentration in the supernatant after hydrothermal reaction.

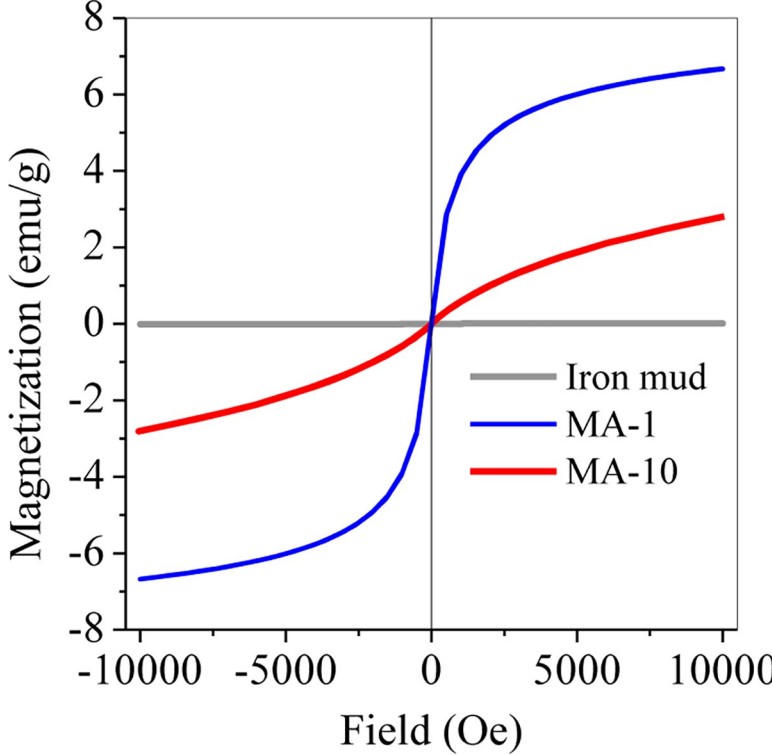

**Fig 6. Magnetisation of the sludge, MA-1 and MA-10.**

MA-10 were fitted with pseudo-first-order and pseudo-second order models, separately. Such parameters are summarised in Table 2. Pseudo-second-order model provided a good description of chromate adsorption on MAs, that is, chemisorption between chromate and MAs was predominant. Moreover, MA-10 showed higher equilibrium adsorption capacity ($q_e$) than MA-1, demonstrating that MA-10 was more effective in chromate adsorption than MA-1.

The adsorption isotherm of chromate on MA-1 and MA-10 were further investigated. The equilibrium data were fitted with both Langmuir and Freundlich models (Fig 9 and Table 2). Compared with the Freundlich model, the Langmuir model fitted well to the adsorption of chromate on MA-1 and MA-10, suggesting that MA-1 and MA-10 had an energetically homogeneous surface for chromate adsorption [28]. The maximum adsorption capacity ($q_m$) of MA-10 was 183.2 mg/g, which was lower than 222.2 mg/g on magnetic graphene oxide [29], but was higher than 51.8 mg/g on jacobsite/chitosan nanocomposites [30], 153.9 mg/g on magnetic chitosan particles [31], and 169.5 mg/g on polypyrrole/$Fe_3O_4$ nanocomposite [32] (Table 3). Magnetic graphene oxide was an expensive man-made carbon material, which should increase the cost of wastewater treatment. On the contrary, MA-10, synthesized using the waste sludge as raw material, which was a low-cost effective adsorbent for chromate adsorption.

### 3.5. Adsorption mechanism of chromate by MA-10

XPS and Mössbauer experiments were performed to investigate the adsorption mechanism of chromate on MA-1 and MA-10. As shown in Fig 9(A), a peak at binding energy of 579.2 eV was observed in MA-1 after adsorption. This peak was attributed to Cr(VI) in chromate [43], indicating that chromate predominated on MA-1 surface. No peak of Cr (III) was observed.

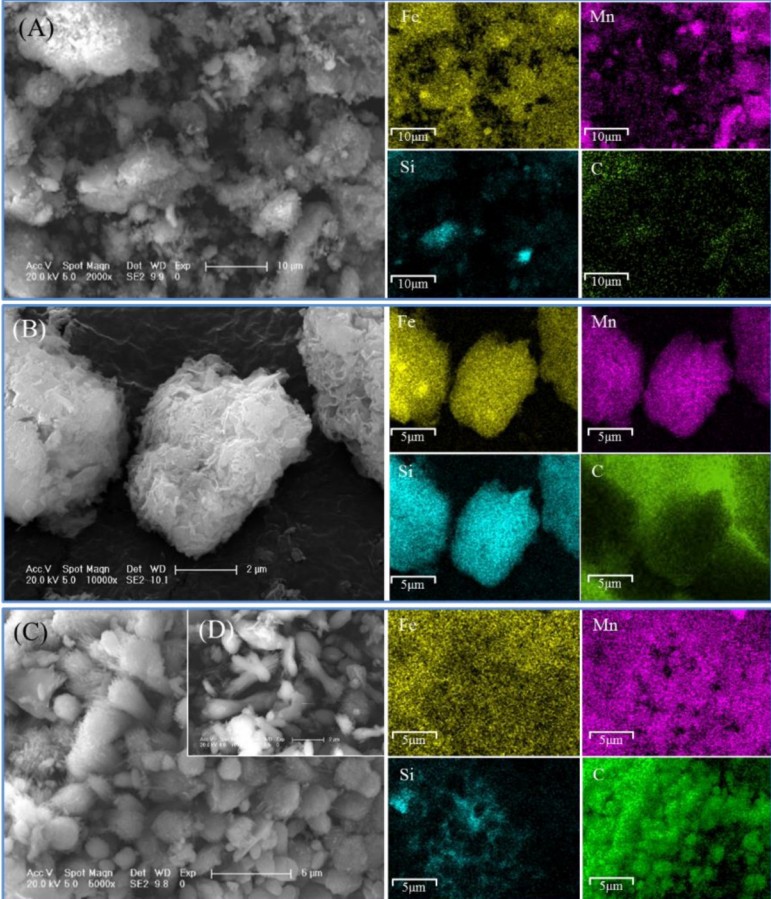

**Fig 7.** SEM pictures of (A) the sludge, (B) MA-1 and (C and D) MA-10.

Therefore, no redox reaction occurred in adsorption. Compared with MA-1, MA-10 showed two peaks at 579.2 and 576.8 eV in XPS spectra (Fig 10(A)). These peaks were affiliated to chromate and $Cr^{3+}$ of Cr-O bond [36]. Hence, chromate and $Cr^{3+}$ were adsorbed on MA-10. After adsorption, only one peak at binding energy of 642 eV affiliated to $Mn^{4+}$ was observed (Fig 10(B)), indicating that $Mn^{2+}$ in rhodochrosite was involved in the reduction of chromate. Mössbauer spectra showed that the relative area decreased by 26.2% for siderite but increased by 25.7% for ferrihydrite in MA-10 (Fig 10(C) and Table 4). Therefore, $Fe^{2+}$ in siderite was oxidised by chromate and further hydrolysed in the form of ferrihydrite.

Chromate, which could oxidise Fe2+/Mn2+-containing compounds, was predominant in the form of $HCrO_4^-$ in acidic solution [36]. When MA-1 was introduced to the acidic solution, its surface functional groups $\equiv$Me-O-H (Me represented Fe, Mn and Si) reacted with chromate via surface coordination with the release of one molecule of $H_2O$ (Eq 1), resulting in chromate adsorption. Jacobsite was a $Mn^{2+}$-containing compound in MA-1 covered with ferrihydrite. Thus, the oxidation of jacobsite by chromate was inhibited. This result agreed well with the no observation of $Cr^{3+}$ on MA-1 surface after adsorption [44]. However, siderite and rhodochrosite were rich in MA-10. They reacted with chromate via redox reaction with generation of $Fe^{3+}/Mn^{4+}$ and $Cr^{3+}$ on MA-10 (Eqs 2 & 3), followed by surface coprecipitation in the form of mixed Fe/Mn-Cr hydroxide (Eqs 4 & 5) [45]. This process predominated the chromate adsorption on MA-10. In addition, similar to MA-1, the newly formed Fe/Mn hydroxide had

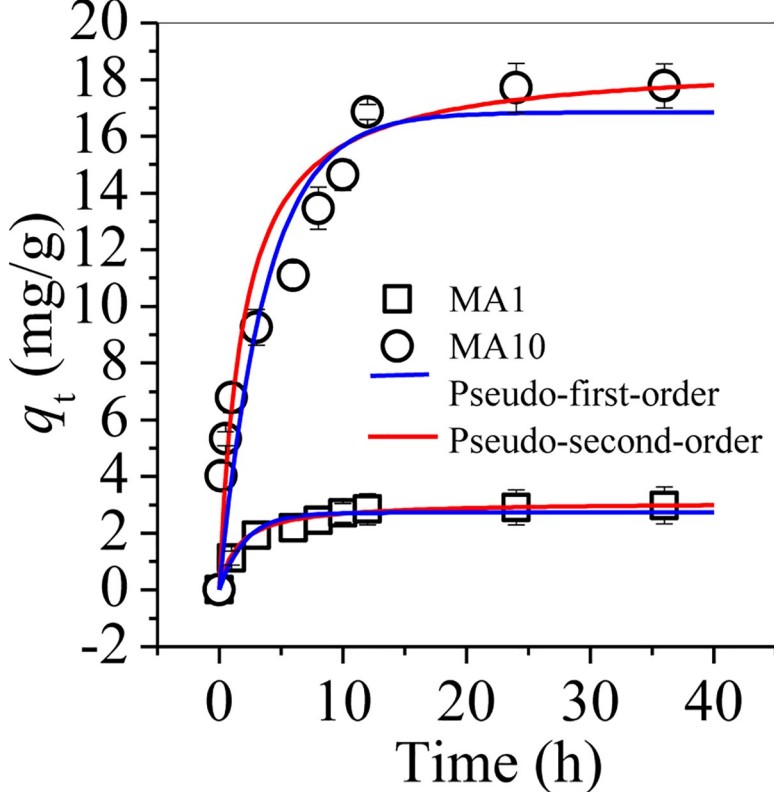

**Fig 8. Adsorption kinetics of chromate adsorption by MA-1 and MA-10.**

abundant hydroxyl groups for chromate coordination (Eq 1). Therefore, a combined effect of redox reaction and surface coordination occurred in chromate adsorption on MA-10. This effect significantly improved the adsorption capacity of MA-10 compared with MA-1.

$$\equiv S{-}O{-}H + HCrO_4^- \rightarrow \equiv S{-}CrO_4^- + H_2O, \qquad (1)$$

$$3FeCO_3 + HCrO_4^- + 10H^+ \rightarrow 3Fe^{3+} + Cr^{3+} + 4H_2O + 3HCO_3^-, \qquad (2)$$

**Table 2. Parameters for chromate adsorption on MA-1 and MA-10.**

| Adsorption models | Parameters | MA-1 | MA-10 |
|---|---|---|---|
| Pseudo-first-order model | $R^2$ | 0.964 | 0.878 |
| | $k_1$ (L/h) | 0.352 | 0.261 |
| | $q_e$(mg/g) | 2.82 | 16.85 |
| Pseudo-second-order model | $R^2$ | 0.99 | 0.988 |
| | $k_2$ ($10^{-3}$ g/mg·h) | 0.212 | 0.028 |
| | $q_e$ (mg/g) | 3.09 | 18.65 |
| Langmuir model | $R^2$ | 0.997 | 0.996 |
| | $q_m$(mg/g) | 21.1 | 183.2 |
| | $K_L$(L/mg) | 0.005 | 0.029 |
| Freundlich model | $R^2$ | 0.971 | 0.96 |
| | $1/n$ | 0.65 | 0.43 |
| | $K_F$((mg/g)(L/mg)$^{1/n}$) | 0.269 | 13.06 |

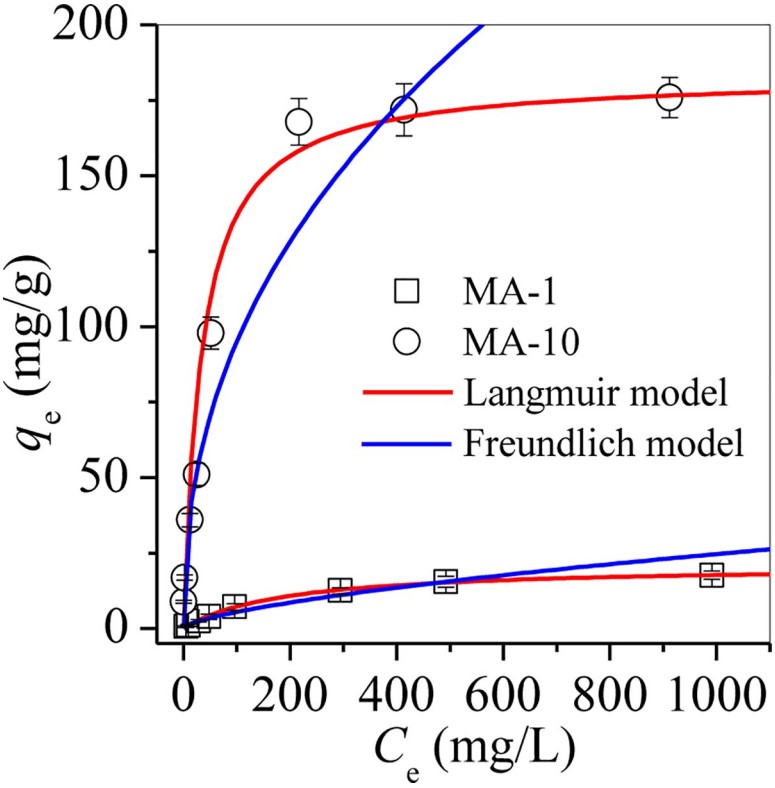

**Fig 9. Adsorption isotherm of chromate on MA-1 and MA-10.**

$$3MnCO_3 + 2HCrO_4^- + 17H^+ \rightarrow 3Mn^{4+} + 2Cr^{3+} + 8H_2O + 3HCO_3^-, \tag{3}$$

$$Cr^{3+} + 3Fe^{3+} + 12H_2O \rightarrow CrFe_3(OH)_{12} + 12H^+, \tag{4}$$

$$2Cr^{3+} + 3Mn^{4+} + 18H_2O \rightarrow Cr_2Mn_3(OH)_{18} + 18H^+, \tag{5}$$

**Table 3. Adsorption capacity of chromate on MA-10 in comparison with the other Fe-containing adsorbent.**

| Synthesised adsorbent | Raw material | pH | $q_m$ (mg/g) | Reference |
|---|---|---|---|---|
| MA-10 | Groundwater treatment sludge | 4 | 183.2 | This work |
| Polypyrrole/$Fe_3O_4$ nanocomposite | Chemical reagent | 2 | 169.5 | [33] |
| Magnetic chitosan particles | Chemical reagent | 4 | 153.9 | [34] |
| Polypyrrole modified montmorillonite | Natural montmorillonite clay | 2 | 119.3 | [35] |
| $Nb_2O_5$ nanorods modified diatomite | Diatomite | 4 | 115 | [36] |
| Magnetic cotton stalk biochar | Iron sludge and cotton stalk biochar | 1.1 | 67.4 | [5] |
| Jacobsite/chitosan nanocomposites | Chemical reagent | 2 | 51.8 | [37] |
| Chitosan modified fly ash | Fly ash | 5 | 33.3 | [6] |
| Surface modified jacobsite | Chemical reagent | 2 | 31.6 | [38] |
| Cetyltrimethylammonium bromide modified red mud | Red mud | 2 | 22.2 | [39] |
| Polypyrrole modified biochar | Red mud | 5.3 | 20.8 | [40] |
| Lanthanum modified red mud | Red mud | 9 | 17.4 | [41] |
| Hexadecyltrimethylammonium bromide modified nanozeolite A | Commercial zeolite A | 3 | 14.2 | [42] |

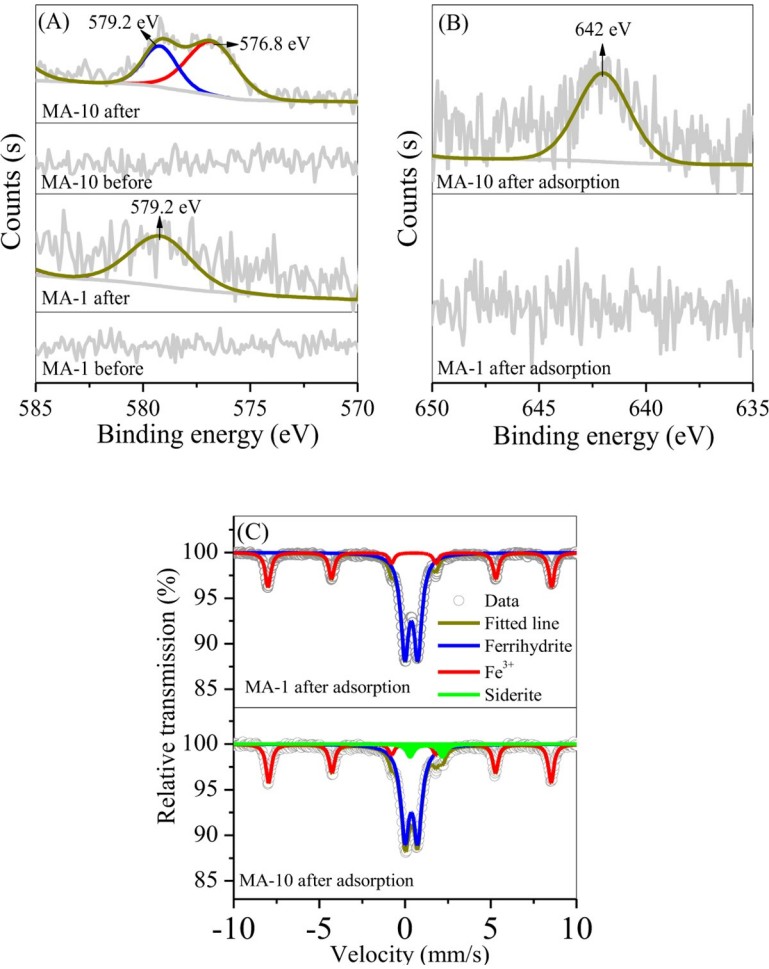

**Fig 10.** High resolution (A) Cr 2p and (B) Mn 2p XPS curves of MAs before and after chromate adsorption and (C) Mössbauer curves of MAs after adsorption.

### 3.6. Nontoxicity of MA-10

MA-10 showed superior adsorption capacity of Cr, and thus it released Fe, Mn, Al and Si in the adsorption process was also determined in accordance the method of Kaur et al. [46]. MA-10 was stable at neutral and alkaline solutions, in which the released Fe, Mn, Al and Si was lower than 0.02 mg/L after leaching for 48 h (Fig 11). But, at acidic solution, the released Fe/Mn were 0.18 and 0.04 mg/L (Fig 11), separately, due to the dissolution of Fe/Mn-bearing compounds (e.g. siderite and rhodochrosite) in MA-10. However, the released Fe/Mn

**Table 4. Mössbauer parameters of MA-1 and MA-10 after chromate adsorption.**

| Sample | Component | Isomer shift (mm/s) | Quadruple split (mm/s) | Hyperfine field (KOe) | Relative absorption area (%) |
|---|---|---|---|---|---|
| MA-1 after adsorption | Ferrihydrite | 0.23 | 0.78 | | 53.9 |
| | $Fe^{3+}$ | 0.26 | 0.23 | 513.3 | 46.1 |
| MA-10 after adsorption | Ferrihydrite | 0.23 | 0.79 | | 60.3 |
| | Siderite | 1.11 | 1.87 | | 5.4 |
| | $Fe^{3+}$ | 0.26 | 0.22 | 511.5 | 34.3 |

concentrations were also meet the discharge standard for smelting wastewater of China [47]. In addition, the concentrations of heavy metals, e.g. Zn, Cu and Pb, were below the detectable limit due to the absence of them in the MA-10 and raw sludge.

The sludge mainly consists of two Fe-bearing minerals, ferrihydrite, and hematite. Ferrihydrite is weakly crystallized and can be easily converted into magnetic species, such as maghemite, with hematite as the final product [48, 49]. The conversion commonly initiated in the absence of reducing reagent, such as ascorbic acid. In our previous study, the impurity Si/Al oxides (quartz and boehmite) were dissolved to $Si(OH)_4^-$ and $Al(OH)_4$ after hydrothermal treatment with 6 M NaOH, and then approximately 24.6% ferrihydrite in the Al/Fe-rich sludge was converted to maghemite [10]. The formation of maghemite conferred good magnetic response on the hydrothermal product. Such magnetic product used many surface hydroxyl groups, such as $\equiv$Fe-OH, $\equiv$Mn-OH, $\equiv$Al-OH, and $\equiv$Si-OH, and had negatively charged surface [13, 48, 49], with high affinity for adsorbing heavy metals (e.g., Cu, Zn, and Ni) [13, 31, 50] and cationic organics (e.g. methylene blue [12], tetracycline, and oxytetracycline [32, 48]). In this study, MA-1 prepared at molar ratio of 1 exhibited similar surface functional groups to these products. However, its adsorption for $HCrO_4^-$ was unsatisfactory because $HCrO_4^-$ was an anion and repelled by the negatively charged MA-1 surface.

The introduction of ascorbic acid in the hydrothermal system served as strong reducer and reacted with Fe/Mn-bearing minerals in the sludge with the generation of magnetic jacobsite $MnFe_2O_4$. Such Fe/Mn-bearing minerals included ferrihydrite, well crystallized hematite, and Mn oxides. Only redox reaction between ascorbic acid and Fe oxides occurred to generate $Fe^{2+}$ [30]. when the Mn oxides were absent. Then, the $Fe^{2+}$ was reoxidized by residual dissolved oxygen in the hydrothermal system [12], to regenerate $Fe^{3+}$ and was involved in the formation of magnetic species in two processes. The first process was the coprecipitation of $Fe^{2+}$ and $Fe^{3+}$ in the form of magnetite [30], and the second process was the hydrolysis of $Fe^{3+}$ to Fe oxyhydroxide and recrystallized in the form of maghemite [12]. Given that several solid wastes, such as red mud [30] and fly ash [51], were rich in Fe/Mn oxides, they can be directly converted to magnetic adsorbents via the hydrothermal method with ascorbic acid.

Such adsorbents were efficient in the removal of cationic heavy metals but unsuitable in the adsorption of anion $HCrO_4^-$. However, the reduction reaction of Fe/Mn-bearing minerals in Fe/Mn-rich waste continued with the addition of adequate ascorbic acid to generate $Fe^{2+}/Mn^{2+}$ in the involvement of siderite/rhodochrosite. This reaction provided a strategy to generate siderite/rhodochrosite on magnetic adsorbent surface. The results showed that the product MA-10, prepared at the molar ratio of 10 showed a high removal capacity of $HCrO_4^-$ [52].

The benefit of recycling groundwater treatment sludge to prepare magnetic adsorbent was twofold. First, the sludge is a typical solid waste and easily converted into a magnetic adsorbent via a one-step hydrothermal method. No exogenous Fe, Si, and Al were added to the hydrothermal process, indicating that the conversion of sludge into the magnetic adsorbent was green and feasible. Second, the obtained magnetic adsorbent, especially MA-10, exhibited a desirable chromate adsorption capacity [53]. It could also adsorb various wastewater contaminants, including Mn [54] and F [55]. These advantages demonstrated that the prepared magnetic adsorbent has potential application in environment pollution control. Future studies should be performed to reduce the cost of magnetic adsorbent synthesis and test the effectiveness of magnetic adsorbent in wastewater treatment.

## 4. Conclusion

Groundwater treatment sludge is composed of Fe/Mn oxides and impurity Si/Al oxides, such as dmisteinbergite and kaolinite. It was converted to magnetic adsorbent via a facile

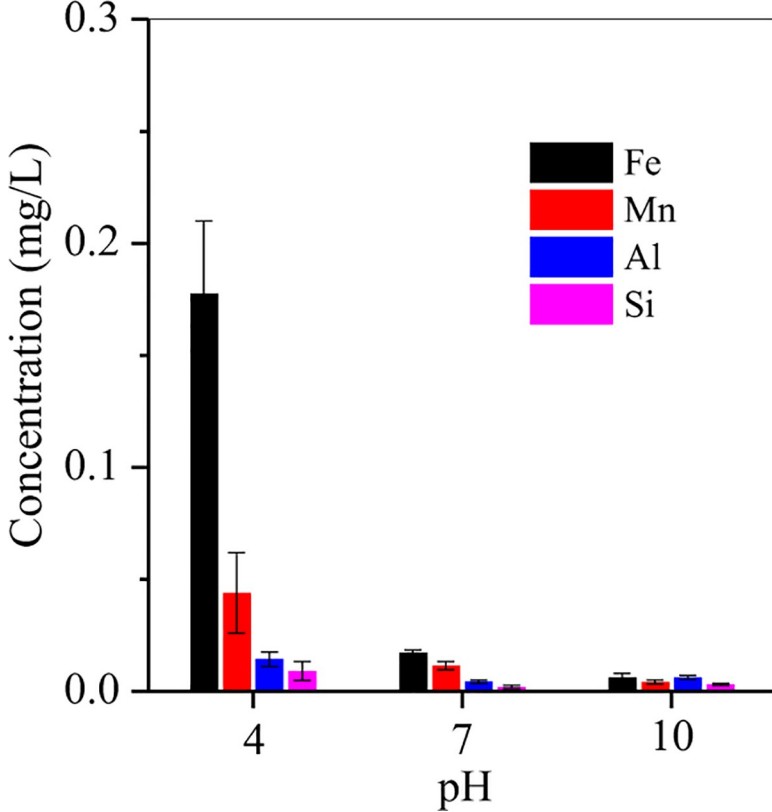

**Fig 11. The release of Fe, Mn Al and Si from MA-10.**

hydrothermal method using ascorbic acid as reducing reagent. Fe and Mn were 28.8 and 8.1 wt.% in the sludge and were involved in the formation of jacobsite, providing the synthesized adsorbent with magnetic property. Such adsorbent was generated in four steps, namely, (1) the oxidation of ascorbic acid by dissolved oxygen to generate carbonate in the solution; (2) the reductive dissolution of Fe/Mn oxides by ascorbic acid to generate $Fe^{2+}$ and $Mn^{2+}$; (3) the reoxidization of $Fe^{2+}$ by Mn oxides in the formation of $MnFe_2O_4$; (4) the carbonate accumulated in the solution and reacted with residual $Fe^{2+}$ and $Mn^{2+}$ to form siderite and rhodochrosite, respectively. The optimal synthesized adsorbent was MA-10 when the molar ratio of ascorbic acid to Fe was 10. It exhibited a good chromate adsorption capacity of 183.2 mg/g, which was higher than MA-1 generated at the molar ratio of 1. The adsorption kinetic of chromate on MA-10 belonged to the pseudo-second-order, and the simulated equilibrium data showed a Langmuir sorption isotherm. Combining the absorption results, the groundwater treatment sludge might be viewed as a satisfactory raw source to prepare magnetic adsorbents with high performance in chromate-bearing wastewater treatment.

## Supporting information

**S1 Data. Graphic picture.**
(DOCX)

**S2 Data. Supplementary related method.**
(DOCX)

## Author Contributions

**Conceptualization:** Zhan Qu, Yu Chen, Suiyi Zhu.

**Data curation:** Zhan Qu.

**Funding acquisition:** Yu Chen, Suiyi Zhu.

**Investigation:** Ge Dong, Suiyi Zhu, Yang Yu.

**Methodology:** Zhan Qu, Yang Yu.

**Project administration:** Suiyi Zhu.

**Resources:** Zhan Qu, Dejun Bian.

**Software:** Wenqing Dong, Yu Chen.

**Supervision:** Wenqing Dong, Yu Chen, Suiyi Zhu.

**Validation:** Suiyi Zhu.

**Visualization:** Yu Chen, Ge Dong.

**Writing – original draft:** Zhan Qu.

**Writing – review & editing:** Zhan Qu, Suiyi Zhu.

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
