## [Decision Letter · Decision Letter 0]

16 Mar 2020

PONE-D-19-34495

Upcycling of groundwater treatment sludge to magnetic Fe/Mn-bearing nanorod for chromate adsorption from wastewater treatment

PLOS ONE

Dear Mr. ZHAN,

Thank you for submitting your manuscript to PLOS ONE. After careful consideration, we feel that it has merit but does not fully meet PLOS ONE’s publication criteria as it currently stands. Therefore, we invite you to submit a revised version of the manuscript that carefully addresses the points raised during the review process.

We would appreciate receiving your revised manuscript by Apr 30 2020 11:59PM. To enhance the reproducibility of your results, we recommend that if applicable you deposit your laboratory protocols in protocols.io, where a protocol can be assigned its own identifier (DOI) such that it can be cited independently in the future. For instructions see: http://journals.plos.org/plosone/s/submission-guidelines#loc-laboratory-protocols

We look forward to receiving your revised manuscript. The suggested references by the referees towards citations are optional and and as per wish of the authors.

Kind regards,

Yogendra Kumar Mishra, Ph. D.

Academic Editor

PLOS ONE

Journal Requirements:

"Conceptualization, Zhan Qu; Data curation, Suiyi Zhu; Investigation, Zhan Qu and Wenqing Dong; Methodology, Yang Yu, Yu chen and Dejun Bian; Validation, Ge Dong; Writing – original draft, Zhan Qu and Suiyi Zhu."

Please provide an amended Funding Statement that declares *all* the funding or sources of support received during this specific study (whether external or internal to your organization) as detailed online in our guide for authors at http://journals.plos.org/plosone/s/submit-now.  Please state what role the funders took in the study.  If any authors received a salary from any of your funders, please state which authors and which funder. If the funders had no role, please state: "The funders had no role in study design, data collection and analysis, decision to publish, or preparation of the manuscript."

5. Please ensure that you refer to Figure "Graphic picture" in your text as, if accepted, production will need this reference to link the reader to the figure.

Reviewers' comments:

Reviewer's Responses to Questions

**Comments to the Author**

1. Is the manuscript technically sound, and do the data support the conclusions?

Reviewer #1: Yes

Reviewer #2: No

2. Has the statistical analysis been performed appropriately and rigorously? 

Reviewer #1: Yes

Reviewer #2: Yes

3. Have the authors made all data underlying the findings in their manuscript fully available?

Reviewer #1: Yes

Reviewer #2: Yes

4. Is the manuscript presented in an intelligible fashion and written in standard English?

Reviewer #1: Yes

Reviewer #2: Yes

5. Review Comments to the Author

Reviewer #1: Presents manuscript by Qu et al. reported the removal of chromate from wastewater using Fe/Mn bearing sludge. Hydrothermal treatment was done for the recrystallization of Fe/Mn using ascorbate. The whole writing and logical flow are clear and straightforward. The manuscript can be accepted for publication after addressing the following comments.

1. Mention if the readings were taken in triplicates in this study?

2. Provide the zeta potential value of the solution before and after hydrothermal treatment to show the nature of functional groups.

3. The discussions on adsorption mechanism and selectivity should be extended to enhance the current manuscript based on some related references: ACS Applied Materials and Interfaces, 2019, 11, 18165-18177. ACS Sustainable Chemistry & Engineering, 2019, 7, 3772−3782. ACS Applied Materials and Interfaces, 2019, 11, 43949-43963.

Reviewer #2: PONE-D-19-34495

Upcycling of groundwater treatment sludge to magnetic Fe/Mn-bearing nanorod for

chromate adsorption from wastewater treatment

Dear Editor,

In this article “Upcycling of groundwater treatment sludge to magnetic Fe/Mn-bearing

nanorod for chromate adsorption from wastewater treatment” the author have investigated chromate adsorption from wastewater using sludge derived magnetic Fe/Mn-bearing nanorods. The subject is interesting and cover the journal aims. Analysis and characterization of the composite may need detail discussion. The current version of the paper is not acceptable and need major revision.

The observations are as follows

1. The novelty of the work should be highlighted.

2. XRF analysis need detail explanation as it is missing in result and discussion part.

3. How the author confirms that the magnetic Fe/Mn-bearing nanorod derived from sludge is not toxic, some toxicity test/leachability test must be conducted to ensure nontoxicity of magnetic Fe/Mn-bearing nanorod in water.

4. In abstract and conclusion, it is mentioned that “This MA-10 showed 183.2 mg/g of chromate adsorption capacity” but in result and discussion the adsorption capacity is not discussed. The detail experiment explaining adsorption capacity must be discussed.

5. The advantage of taking magnetic adsorbent for chromate adsorption must be added in the introduction.

6. Table 2 shows Adsorption capacity of chromate by MA-10 is highest at pH2 but in section 2.3 the pH is mentioned 4.

7. Fig. 6 the scale of SEM images is not clear. How the author claims the composition of

Fe/Mn nanorods.

8. The results have not been explored in enough depth. Comprehensive discussion of the findings is completely missing from the result discussion section.

9. The conclusion should be in more detail.

6. PLOS authors have the option to publish the peer review history of their article (what does this mean?). If published, this will include your full peer review and any attached files.

Reviewer #1: No

Reviewer #2: No

---

## [Author Response · Author response to Decision Letter 0]

9 Apr 2020

Response to Journal Requirements:

Response: Thank you for your suggestion. The manuscript was reformatted in accordance with the journal requirement. 

Response: Thank you for your suggestion. The permission information was added to line 321-323 on page 16.

"Conceptualization, Zhan Qu; Data curation, Suiyi Zhu; Investigation, Zhan Qu and Wenqing Dong; Methodology, Yang Yu, Yu chen and Dejun Bian; Validation, Ge Dong; Writing – original draft, Zhan Qu and Suiyi Zhu."

Please provide an amended Funding Statement that declares *all* the funding or sources of support received during this specific study (whether external or internal to your organization) as detailed online in our guide for authors at http://journals.plos.org/plosone/s/submit-now. 

Please state what role the funders took in the study. If any authors received a salary from any of your funders, please state which authors and which funder. If the funders had no role, please state: "The funders had no role in study design, data collection and analysis, decision to publish, or preparation of the manuscript."

Response: Thank you for your suggestion. The corresponding statement was added to line 317-319 on page 13.

Response: Thank you for your suggestion. My ORCID ID was added.

5. Please ensure that you refer to Figure "Graphic picture" in your text as, if accepted, production will need this reference to link the reader to the figure.

Response: Thank you for your suggestion. The graphic picture was cited in the manuscript as shown in line 9 on page 1.

Response: The description of supporting information files was added at the end of manuscript, as shown in line 314 - 316 on page 16. 

Response to Reviewers' comments:

Reviewer #1: Presents manuscript by Qu et al. reported the removal of chromate from wastewater using Fe/Mn bearing sludge. Hydrothermal treatment was done for the recrystallization of Fe/Mn using ascorbate. The whole writing and logical flow are clear and straightforward. The manuscript can be accepted for publication after addressing the following comments.

1. Mention if the readings were taken in triplicates in this study?

Response: Thank you for your suggestion. The related description was added to line 80 on page 3.

Each experiment was performed in triple, and average data were reported.

2. Provide the zeta potential value of the solution before and after hydrothermal treatment to show the nature of functional groups.

Response: Thank you for your suggestion. The related description has been added to line 106-109 on page 4.

At the same time, zeta potential test was conducted on the original iron mud and hydrothermal reaction products. The results showed that the zeta potential of the original iron mud changes from 7.5 mV to −18.5 mV (MA-1) and −39.6 mV (MA-10), thereby proving that the surface of hydrothermal reaction product has negative charge. In an aqueous system, the surface of ferrihydrite is covered with -FeOH groups[1].

3. The discussions on adsorption mechanism and selectivity should be extended to enhance the current manuscript based on some related references: ACS Applied Materials and Interfaces, 2019, 11, 18165-18177. ACS Sustainable Chemistry & Engineering, 2019, 7, 3772−3782. ACS Applied Materials and Interfaces, 2019, 11, 43949-43963.

Response: Thank you for your suggestion. The adsorption performance of Cr on the prepared adsorbents have been discussed. Such valuable references are helpful in improving our manuscript and have been added to line 236, 287 and 292.

Reviewer #2: In this article “Upcycling of groundwater treatment sludge to magnetic Fe/Mn-bearing

nanorod for chromate adsorption from wastewater treatment”, the author has investigated chromate adsorption from wastewater using sludge derived magnetic Fe/Mn-bearing nanorods. The subject is interesting and cover the journal aims. Analysis and characterization of the composite may need detail discussion. The current version of the paper is not acceptable and need major revision.

The observations are as follows

1. The novelty of the work should be highlighted.

Response: Thank you for your comments. The highlights have been added and discussed in the introduction. The related description has been listed to line 33-37 on page 1-2.

2. XRF analysis need detail explanation as it is missing in result and discussion part.

Response: Thank you for your suggestion. 

The composition of sludge, MA-1, and MA-10 was determined through X-ray fluorescence spectroscopy (S4-Explorer, Bruker, XRF, Germany). After hydrothermal treatment, the product MA-1 prepared at Mascorbate/MFe molar ratio (short for molar ratio) of 1 showed a high Fe/Mn content (34.2% and 9.6%, respectively) and a low Si/Al content (4.5% and 1.1, respectively) (Fig. R2) compared with the raw sludge; this finding is because of the dissolution of Si/Al oxides (e.g., kaolinite) under alkaline condition (Fig. 4A) with the release of Si(OH)4− (Fig. 4B) and Al(OH)4 to the solution [2]. However, the Fe and Mn in product MA-10 were 25.4 and 7.1 wt.% (Fig. R2) when the molar ratio increased to 10, which were apparently lower than those in the raw sludge and MA-1 and were assigned to the reductive dissolution of Fe/Mn at neutral condition (Figs. 4A & B). The Si and Al in MA-10 were 10.8 and 3.9 wt.% higher than those in the raw sludge and MA-1, demonstrating that the release of Si/Al to solution was retarded with the solution pH decreasing from 12.1 to 7 (Fig. 4A). The related description has been listed to line 88 - 97 on page 3.

Fig. R2 Relative percentage of Fe, Mn, Si and Al in the sludge, MA-1 and MA-10.

3. How the author confirms that the magnetic Fe/Mn-bearing nanorod derived from sludge is not toxic, some toxicity test/leachability test must be conducted to ensure nontoxicity of magnetic Fe/Mn-bearing nanorod in water.

Response: Thank you for your suggestion. MA-10 showed superior adsorption capacity of Cr and released Fe, Mn, Al, and Si in the adsorption in accordance with the method of Kaur et al. [3]. 

MA-10 was stable at neutral and alkaline solutions, where the released Fe, Mn, Al, and Si was lower than 0.02 mg/L after leaching for 48 h (Fig. R3). At acidic solution, the released Fe/Mn were 0.18 and 0.04 mg/L (Fig. R3), respectively, because of the dissolution of Fe/Mn-bearing compounds (e.g., siderite and rhodochrosite) in MA-10. However, the released Fe/Mn concentrations met the discharge standard for smelting wastewater in China [4]. The concentrations of heavy metals, such as Zn, Cu, and Pb, were below the detectable limit because of their absence in the MA-10 and raw sludge. Such description has been added to line 250-257 on page 11.

Fig. R3 the release of Fe, Mn Al and Si from MA-10. 

4. In abstract and conclusion, it is mentioned that “This MA-10 showed 183.2 mg/g of chromate adsorption capacity” but in result and discussion the adsorption capacity is not discussed. The detail experiment explaining adsorption capacity must be discussed.

Response: Thank you for your suggestion. The discussion of chromate adsorption capacity has been added to line 199-205 on page 8 to 9.

The maximum adsorption capacity (qm) of MA-10 was 183.2 mg/g, which was lower than 222.2 mg/g on magnetic graphene oxide [5], but was higher than 51.8 mg/g on jacobsite/chitosan nanocomposites [6], 153.9 mg/g on magnetic chitosan particles [7], and 169.5 mg/g on polypyrrole/Fe3O4 nanocomposite [8] (Table 3 in manuscript). Magnetic graphene oxide is an expensive man-made carbon material that increases the cost of wastewater treatment. On the contrary, MA-10 synthesized using the waste sludge as raw material is a low-cost effective adsorbent for chromate adsorption.

5. The advantage of taking magnetic adsorbent for chromate adsorption must be added in the introduction.

Response: Thank you for your suggestion. The advantage of magnetic adsorbents in chromate-bearing wastewater treatment was discussed in the introduction, as shown in line 33-37 on page1.

6. Table 2 shows Adsorption capacity of chromate by MA-10 is highest at pH 2 but in section 2.3 the pH is mentioned 4.

Response: Thank you for your suggestion. The adsorption experiment was performed at pH 4. Table 2 has been corrected, as shown on page 1.

7. Fig. 6 the scale of SEM images is not clear. How the author claims the composition of Fe/Mn nanorods.

Response: Thank you for your suggestion. The composition of products MA-1 and MA-10 was characterized through Mössbauer spectroscopy (MP500, Oxford, UK) and XRF, respectively.

For Mössbauer spectroscopy, the structural Fe of Fe-bearing compounds in total Fe in the sludge and products (e.g., MA-1 and MA-10) was detected, and its diffraction intensity was recorded in Mössbauer spectra. After multipeak resolution of Mössbauer spectra on MossWinn 4.0 software, the relative percentages of structural Fe of Fe-bearing compounds in total Fe (short for relative percentage) were calculated from the relative adsorption areas of the subspectra (Table 1). 

For XRF analysis, the relative weight percentage of elements (e.g., Fe, Mn, Al, and Si) in the sludge, MA-1, and MA-10 was characterized, as shown in the response to question 2 in accordance with your suggestion. 

8. The results have not been explored in enough depth. Comprehensive discussion of the findings is completely missing from the result discussion section.

Response: Thank you for your suggestion. The discussion has been added as shown in line 261-295 on page12.

The sludge mainly consists of two Fe-bearing minerals, ferrihydrite, and hematite. Ferrihydrite is weakly crystallized and can be easily converted into magnetic species, such as maghemite, with hematite as the final product [9, 10]. The conversion commonly initiated in the absence of reducing reagent, such as ascorbic acid. In our previous study, the impurity Si/Al oxides (quartz and boehmite) were dissolved to Si(OH)4− and Al(OH)4 after hydrothermal treatment with 6 M NaOH, and then approximately 24.6% ferrihydrite in the Al/Fe-rich sludge was converted to maghemite [11]. The formation of maghemite conferred good magnetic response on the hydrothermal product. Such magnetic product used many surface hydroxyl groups, such as ºFe-OH, ºMn-OH, ºAl-OH, and ºSi-OH, and had negatively charged surface [9, 11-13], with high affinity for adsorbing heavy metals (e.g., Cu, Zn, and Ni) [12, 14, 15] and cationic organics (e.g., methylene blue [13], tetracycline, and oxytetracycline [9, 10]). In this study, MA-1 prepared at molar ratio of 1 exhibited similar surface functional groups to these products. However, its adsorption for HCrO4− was unsatisfactory because HCrO4− was an anion and repelled by the negatively charged MA-1 surface.

The introduction of ascorbic acid in the hydrothermal system served as strong reducer and reacted with Fe/Mn-bearing minerals in the sludge with the generation of magnetic jacobsite MnFe2O4. Such Fe/Mn-bearing minerals included ferrihydrite, well crystallized hematite, and Mn oxides. Only redox reaction between ascorbic acid and Fe oxides occurred to generate Fe2+ [14, 16] when the Mn oxides were absent. Then, the Fe2+ was reoxidized by residual dissolved oxygen in the hydrothermal system [14], to regenerate Fe3+ and was involved in the formation of magnetic species in two processes. The first process was the coprecipitation of Fe2+ and Fe3+ in the form of magnetite [16], and the second process was the hydrolysis of Fe3+ to Fe oxyhydroxide and recrystallized in the form of maghemite [14]. Given that several solid wastes, such as red mud[16] and fly ash [17], were rich in Fe/Mn oxides, they can be directly converted to magnetic adsorbents via the hydrothermal method with ascorbic acid. 

Such adsorbents were efficient in the removal of cationic heavy metals but unsuitable in the adsorption of anion HCrO4−. However, the reduction reaction of Fe/Mn-bearing minerals in Fe/Mn-rich waste continued with the addition of adequate ascorbic acid to generate Fe2+/Mn2+ in the involvement of siderite/rhodochrosite. This reaction provided a strategy to generate siderite/rhodochrosite on magnetic adsorbent surface. The results showed that the product MA-10,prepared at the molar ratio of 10 showed a high removal capacity of HCrO4−. The benefit of recycling groundwater treatment sludge to prepare magnetic adsorbent was twofold. First, the sludge is a typical solid waste and easily converted into a magnetic adsorbent via a one-step hydrothermal method. No exogenous Fe, Si, and Al were added to the hydrothermal process, indicating that the conversion of sludge into the magnetic adsorbent was green and feasible. Second, the obtained magnetic adsorbent, especially MA-10, exhibited a desirable chromate adsorption capacity. It could also adsorb various wastewater contaminants, including Mn 18] and F [19]. These advantages demonstrated that the prepared magnetic adsorbent has potential application in environment pollution control. Future studies should be performed to reduce the cost of magnetic adsorbent synthesis and test the effectiveness of magnetic adsorbent in wastewater treatment.

9. The conclusion should be in more detail.

Response: Thank you for your suggestion. The conclusion has been reorganized as follows. 

Groundwater treatment sludge is composed of Fe/Mn oxides and impurity Si/Al oxides, such as dmisteinbergite and kaolinite. It was converted to magnetic adsorbent via a facile hydrothermal method using ascorbic acid as reducing reagent. Fe and Mn were 28.8 and 8.1 wt.% in the sludge and were involved in the formation of jacobsite, providing the synthesized adsorbent with magnetic property. Such adsorbent was generated in four steps, namely, (1) the oxidation of ascorbic acid by dissolved oxygen to generate carbonate in the solution; (2) the reductive dissolution of Fe/Mn oxides by ascorbic acid to generate Fe2+ and Mn2+; (3) the reoxidization of Fe2+ by Mn oxides in the formation of MnFe2O4; (4) the carbonate accumulated in the solution and reacted with residual Fe2+ and Mn2+ to form siderite and rhodochrosite, respectively. The optimal synthesized adsorbent was MA-10 when the molar ratio of ascorbic acid to Fe was 10. It exhibited a good chromate adsorption capacity of 183.2 mg/g, which was higher than MA-1 generated at the molar ratio of 1. The adsorption kinetic of chromate on MA-10 belonged to the pseudo-second-order, and the simulated equilibrium data showed a Langmuir sorption isotherm. Combining the absorption results, the groundwater treatment sludge might be viewed as a satisfactory raw source to prepare magnetic adsorbents with high performance in chromate-bearing wastewater treatment. 

References

1. Ahmed MA, Ali SM, El-Dek SI, Galal. Magnetite-hematite nanoparticles prepared by green methods for heavy metal ions removal from water. Materials Science & Engineering B. 2013; 178(10): p. 744-751.

2. Ban T, Ohwaki T, Ohya Y, Takahashi Y. Influence of the addition of alkanolamines and tetramethylammonium hydroxide on the shape and size of zeolite-A particles. International Journal of Inorganic Materials. 1999; 1(3): p. 243-251.

3. Kaur N, Singh B, Kennedy BJ. Dissolution of Cr, Zn, Cd, and Pb single- and multi-metal-substituted goethite: relationship to structural, morphological, and dehydroxylation properties. Clays and clay minerals. 2010; 58(3), P.415-430.

 4. Wei Z, Yunping T, Xianqiang Z, Yuqiang L, Guofu H, Yunxia D, et al. Study on the standard discharge technology of mixed wastewater in the area of Chemical Industrial Park. Industrial Water Treatment. 2010.

5. Alizadeh A. Graphene oxide/Fe3O4/SO3H nanohybrid: a new adsorbent for adsorption and reduction of Cr (VI) from aqueous solutions. RSC advances. 2017; 7(24): p. 14876-14887.

6. Xiao Y, Liang H, Wang Z. MnFe2O4/chitosan nanocomposites as a recyclable adsorbent for the removal of hexavalent chromium. Materials Research Bulletin. 2013; 48(10): p. 3910-3915.

7. Chaofan Z, Huaili Z, Yongjuan W, Yili W, Wenqi Q, Qiang A, et al. Synthesis of novel modified magnetic chitosan particles and their adsorption performance toward Cr(VI). Bioresour Technol. 2018; 267: p. 1-8.

8. Bhaumik M, Maity A, Srinivasu VV,Onyango MS. Enhanced removal of Cr(VI) from aqueous solution using polypyrrole/Fe3O4 magnetic nanocomposite. Journal of Hazardous Materials. 2011; 190(1): p. 381-390.

9. Qu Z, Dong G, Zhu S, Yu Y, Huo M. Recycling of groundwater treatment sludge to prepare nano-rod erdite particles for tetracycline adsorption. Journal of Cleaner Production. 2020; 257: p. 120462.

10. Qu Z, Wu Y, Zhu S, Yu Y, Huo M, Zhang L, et al. Green Synthesis of Magnetic Adsorbent Using Groundwater Treatment Sludge for Tetracycline Adsorption. Engineering, 2019. 5(5): p. 880-887.

11. Zhu S, Wu Y, Qu Z, Zhang L, Yu Y, Xie X, et al. Green synthesis of magnetic sodalite sphere by using groundwater treatment sludge for tetracycline adsorption. Journal of Cleaner Production. 2020; 247: p. 119140.

12. Ong DC, Kan CC, Pingul-Ong S MB, De Luna MDG. Utilization of groundwater treatment plant (GWTP) sludge for nickel removal from aqueous solutions: isotherm and kinetic studies. Journal of Environmental Chemical Engineering. 2017; 5(6): p. 5746-5753.

13. Zhu S, Fang S, Huo M, Yu Y, Chen Y, Yang X, et al. A novel conversion of the groundwater treatment sludge to magnetic particles for the adsorption of methylene blue. Journal of Hazardous Materials. 2015; 292: p. 173-179.

14. Zhu S, Dong G, Yu Y, Yang J, Yang W, Fan W, et al. Hydrothermal synthesis of a magnetic adsorbent from wasted iron mud for effective removal of heavy metals from smelting wastewater. Environmental Science and Pollution Research. 2018; 25(23): p. 22710-22724.

15. Zhu S, Dong G, Lin x. Valorization of manganese-containing groundwater treatment sludge by preparing magnetic adsorbent for Cu (II) adsorption. Journal of environmental management. 2019; 236: p. 446-454.

16. Dong W, Liang K, Qin Y, Ma H, Zhao X, Zhang L, et al. Hydrothermal Conversion of Red Mud into Magnetic Adsorbent for Effective Adsorption of Zn (II) in Water. Applied Sciences. 2019; 9(8): p. 1519.

17. Wang S, Boyjoo Y, Choueib A, Zhu ZH. Removal of dyes from aqueous solution using fly ash and red mud. Water Research. 2005; 39(1): p. 129.

18. Paul, Wersin, Laurent, Charlet. From adsorption to precipitation: Sorption of Mn2+ on FeCO3(s). Geochimica Et Cosmochimica Acta. 1989.

19. Zhang Y, Jia Y. Fluoride adsorption on manganese carbonate: Ion-exchange based on the surface carbonate-like groups and hydroxyl groups. Journal of Colloid & Interface Science. 2017; p. S0021979717311219.

---

## [Decision Letter · Decision Letter 1]

20 May 2020

Upcycling of groundwater treatment sludge to magnetic Fe/Mn-bearing nanorod for chromate adsorption from wastewater treatment

PONE-D-19-34495R1

Dear Dr. ZHAN,

We are pleased to inform you that your manuscript has been judged scientifically suitable for publication and will be formally accepted for publication once it complies with all outstanding technical requirements.

With kind regards,

Yogendra Kumar Mishra, Ph. D.

Academic Editor

PLOS ONE

Additional Editor Comments (optional):

Reviewers' comments:

Reviewer's Responses to Questions

**Comments to the Author**

1. If the authors have adequately addressed your comments raised in a previous round of review and you feel that this manuscript is now acceptable for publication, you may indicate that here to bypass the “Comments to the Author” section, enter your conflict of interest statement in the “Confidential to Editor” section, and submit your "Accept" recommendation.

Reviewer #1: All comments have been addressed

Reviewer #2: All comments have been addressed

2. Is the manuscript technically sound, and do the data support the conclusions?

Reviewer #1: Yes

Reviewer #2: Yes

3. Has the statistical analysis been performed appropriately and rigorously? 

Reviewer #1: Yes

Reviewer #2: N/A

4. Have the authors made all data underlying the findings in their manuscript fully available?

Reviewer #1: Yes

Reviewer #2: Yes

5. Is the manuscript presented in an intelligible fashion and written in standard English?

Reviewer #1: Yes

Reviewer #2: Yes

6. Review Comments to the Author

Reviewer #1: (No Response)

Reviewer #2: All the previous comments have been answered into modified version of the manuscript. The reviewed manuscript meets the criteria and in my opinion can be published as original paper in PLOS ONE journal.

7. PLOS authors have the option to publish the peer review history of their article (what does this mean?). If published, this will include your full peer review and any attached files.

Reviewer #1: No

Reviewer #2: No

---

## [Editor Report · Acceptance letter]

26 May 2020

PONE-D-19-34495R1 

Upcycling of groundwater treatment sludge to magnetic Fe/Mn-bearing nanorod for chromate adsorption from wastewater treatment 

Dear Dr. Qu:

I am pleased to inform you that your manuscript has been deemed suitable for publication in PLOS ONE. Congratulations! Your manuscript is now with our production department. 

With kind regards,

on behalf of

Professor Yogendra Kumar Mishra 

Academic Editor

PLOS ONE